# The Sonophotocatalytic Degradation of Pharmaceuticals in Water by MnOx-TiO2 Systems with Tuned Band-Gaps

**Zahra Khani** [1], **Dalma Schieppati** [1], **Claudia L. Bianchi** [2]  and **Daria C. Boffito** [1,*]

[1] Chemical Engineering Department, Polytechnique Montréal, C.P. 6079, Succ. CV, Montréal, QC H3C 3A7, Canada; zahra-3.khani@polymtl.ca (Z.K.); dalma.schieppati@polymtl.ca (D.S.)

[2] Dipartimento di Chimica, Università degli Studi di Milano, via Golgi 19, 20133 Milano, Italy; claudia.bianchi@unimi.it

[*] Correspondence: daria-camilla.boffito@polymtl.ca

**Abstract:** Advanced oxidation processes (AOPs) are technologies to degrade organic pollutants to carbon dioxide and water with an eco-friendly approach to form reactive hydroxyl radicals. Photocatalysis is an AOP whereby $TiO_2$ is the most adopted photocatalyst. However, $TiO_2$ features a wide (3.2 eV) and fast electron-hole recombination. When Mn is embedded in $TiO_2$, it shifts the absorption wavelength towards the visible region of light, making it active for natural light applications. We present a systematic study of how the textural and optical properties of Mn-doped $TiO_2$ vary with ultrasound applied during synthesis. We varied ultrasound power, pulse length, and power density (by changing the amount of solvent). Ultrasound produced mesoporous $MnO_x$-$TiO_2$ powders with a higher surface area (101–158 $m^2$ $g^{-1}$), pore volume (0-13–0.29 cc $g^{-1}$), and smaller particle size (4–10 μm) than those obtained with a conventional sol-gel method (48–129 $m^2$ $g^{-1}$, 0.14–0.21 cc $g^{-1}$, 181 μm, respectively). Surprisingly, the catalysts obtained with ultrasound had a content of brookite that was at least 28%, while the traditional sol-gel samples only had 7%. The samples synthesized with ultrasound had a wider distribution of the band-gaps, in the 1.6–1.91 eV range, while traditional ones ranged from 1.72 eV to 1.8 eV. We tested activity in the sonophotocatalytic degradation of two model pollutants (amoxicillin and acetaminophen). The catalysts synthesized with ultrasound were up to 50% more active than the traditional samples.

**Keywords:** ultrasound; photocatalysis; sol-gel; $TiO_2$; Mn oxides; band-gap

## 1. Introduction

Water quality is a gargantuan socio-economic issue [1,2]. Improper disposal of drugs, illicit discharge during manufacturing, and the direct or indirect releases by humans and animals are the leading causes of water contamination [3,4]. Micropollutants damage the ecosystem even in the ng $L^{-1}$ to μg $L^{-1}$ range, which is enough to interfere with the endocrine systems of complex organisms and to induce their microbiological resistance. Moreover, these small concentrations accumulate in soil and plants, increasing the local concentration above the μm range [3,5,6]. Many governmental and non-governmental organizations, including the European Union (EU), the World Health Organization (WHO), and the International Program of Chemical Safety (IPCS), provide rules and legal frameworks to protect and improve the quality of the water resources and investigate the long-term effects of these contaminants, which still need further investigation [5]. Antibiotics, anti-inflammatories, hormones, blood lipid regulators, analgesics, and many other recalcitrant pharmaceuticals pollute water in the g $L^{-1}$ orders of magnitude [7]. Acetaminophen (APAP) and amoxicillin (AMO) are the most prescribed

analgesics and antibiotics worldwide, respectively [4,8]. APAP's concentration in several treatment plant effluents is higher than 200 $\mu$g L$^{-1}$, which is sufficient to cause liver-related pathologies and other severe ailments [9–12]. Moreover, antibiotic consumption is expected to grow by 67% in the next five years [3].

Various techniques are currently available to degrade organic contaminants. Specifically, advanced oxidation processes (AOPs) mineralize a variety of organics into $CO_2$, $H_2O$, and/or mineral acids [13,14]. However, their application at the commercial scale is limited by efficiencies, which are still low, and the negative techno-economic assessment. AOPs remove pollutants through the generation of highly oxidative radical species; namely, hydroxyl (HO$^\bullet$), superoxide (O$_2$$^{\bullet-}$), and perhydroxyl (HOO$^\bullet$) radicals. AOPs include (1) ozonation, (2) photocatalysis, and (3) ultrasonic cavitation [13]. In addition, the combination of UV radiation with ultrasonication increases the efficiency of removing contaminants from water [14].

One of the major advantages of photocatalysis over other AOPs is the generation of renewable oxidant sources ($H_2O$ and/or $O_2$ from air), as other AOPs use consumable oxidants [12,15]. $TiO_2$ is one the most employed semiconductors. It has found applications in water and air purification [16–18], sterilization [19], self-cleaning building materials [20], defogging [21], and $H_2$ generation from water splitting [22]. $TiO_2$ is cheaper and more photo-chemically and chemically stable than other semiconductors [13]. Under UV irradiation, $TiO_2$ degrades organic dyes and pesticides in a few hours [14,23]. However, its large band-gap (from 3.0 eV to 3.2 eV), low quantum yield, and fast recombination of electron-hole pairs limits its commercial application under visible light. Metal, metal oxide, and non-metal doping extends the absorption of $TiO_2$ to a longer wavelength [24–30]. The efforts of researchers in the field of photocatalysis aim mainly at extending the band-gap to the visible light region of the solar spectrum ($\lambda > 400$ nm). To cope with the lengthy catalyst preparation steps and to improve the mixing between active phase and dopant(s), spray drying and ultrasound is an approach that has been recently adopted by the authors of this paper [29–34]. The shape and size of catalysts vary with the operating parameters, including the ultrasound irradiation time, temperature, power density, the ultrasonic source (bath-type and horn-type), magnetic stirring, and reactor shape and size [24,35–37]. Conventional sol-gel synthesis procedures require a long processing time. Ultrasound is a mechanical wave that propagates through a succession of compression and rarefaction cycles. It results in the formation of vapor-filled voids that grow and collapse violently to generate hotspots wherein the temperature and pressure reach 5000 K and 20 MPa. The application of ultrasound lowers the process's temperature yet facilitates the transition from an amorphous to a crystalline structure [35]. Acoustic cavitation increases the surface area, controls the particle size distribution, and improves photocatalytic activity [32]. The optimization of the sonication parameters is obviously essential to generating particles with specific characteristics [38]. Stucchi et al. investigated the effect of ultrasonication on Ag-doped $TiO_2$'s photocatalytic degradation of acetone under UV and visible light [32]. Stucchi et al. also report that doping $TiO_2$ with Mn shifts the absorption towards the visible light region of the electromagnetic spectrum [29]. Among all the 3d metals, only Mn absorbs radiation in the visible region and infrared solar light [39,40]. As a consequence, doping $TiO_2$ with Mn reduces the band-gap [40] and may increase the crystallinity of the material [41]. Metals and metal oxides form new electronic states between the valence and conduction bands of $TiO_2$, which reduces the electron-hole recombination rate by acting as an electron trap. This is due to the interaction between the 3d orbitals of Ti and the d orbitals of Mn. Therefore, the distance of charge transfer between electrons of the Mn ions and the conduction or valence band of $TiO_2$ is shortened and the electrons are scavenged more efficiently from the holes, even though the intra-band-gap introduced is narrower [42]. Moreover, the intra-gap creates new electronic states in the $TiO_2$ band-gap, which promotes the d electron transfer from Mn to the conduction band of $TiO_2$ [41].

Neppolian et al. [43] prepared nano-$TiO_2$ photocatalysts using sol-gel and ultrasound-assisted sol-gel methods. They investigated the effect of ultrasonic irradiation time, power density, ultrasonic source, magnetic stirring, initial temperature, and geometry on the reactor. Li et al. [23] combined

ultrasonic cavitation and the hydrothermal method to prepare Fe-doped $TiO_2$ for the photo-degradation of methyl orange (MO). The high crystallinity, large surface area, and larger pore size of the samples prepared photodegraded MO 2.5 times faster. Prasad et al. [37] proved that ultrasound accelerated the synthesis and minimized the agglomeration of $ZrO_2$.

The originality of this work relates to the synthesis of Mn-doped $TiO_2$ catalysts by an ultrasound-assisted sol-gel method, and quantified the effects of power, pulse, and solvent amount on textural and optical properties, including specific surface area (SSA), pore volume, pore size, particle-size distribution, phase composition, and band-gap (eV). XRD, BET, PSD, and UV-spectrophotometry characterized our samples. We compared the physicochemical properties of the samples obtained with ultrasound-assisted sol-gel and a conventional sol-gel process. We also tested their activities by degrading acetaminophen and amoxicillin in a sonophotocatalytic process and correlated the activity with the catalyst features.

## 2. Results and Discussion

### 2.1. X-Ray Diffraction (XRD)

The XRD patterns of the Mn-doped $TiO_2$ calcined at 450 °C and obtained with ultrasound (Figure 1) match with anatase (most intense peak at 25.4°), rutile (most intense peak at 27.4°), and brookite. The (121) brookite peak at 30.8° is unmistakable. The most intense peak of anatase at 25.4° overlaps with the (120) and (111) peaks of brookite at (25.34° and 25.69°) (Table 1).

None of the diffractograms exhibit peaks related to Mn ($MnO_x$), which may mean that Mn is partly well dispersed on the surface and/or dispersed within the $TiO_2$ lattice [44,45]. Indeed, the ionic radius and the charge of the dopant alter the structure of $TiO_2$. If the dopant charge is lower than that of $Ti^{4+}$, it alters the concentration of oxygen vacancies depending on the position within the $TiO_2$ matrix; it either replaces Ti in the lattice or occupies an interstitial position depending on its size and concentration. $Mn^{2+}$ has a larger ionic radius (0.8 Å) than Ti (0.68 Å). Therefore, $Mn^{2+}$ ions only replace $Ti^{4+}$ in the lattice sites. The replacement by metal ions with a valence lower than 4+ and ionic radius higher than 0.68 Å induces oxygen vacancies at the boundaries of anatase grains, which favors bond rupture and solid-state ionic rearrangement [42,44,46,47]. The formation of crystalline phases and their transformation from anatase to rutile depends on the starting material, deposition method, and calcination temperature [45,48,49].

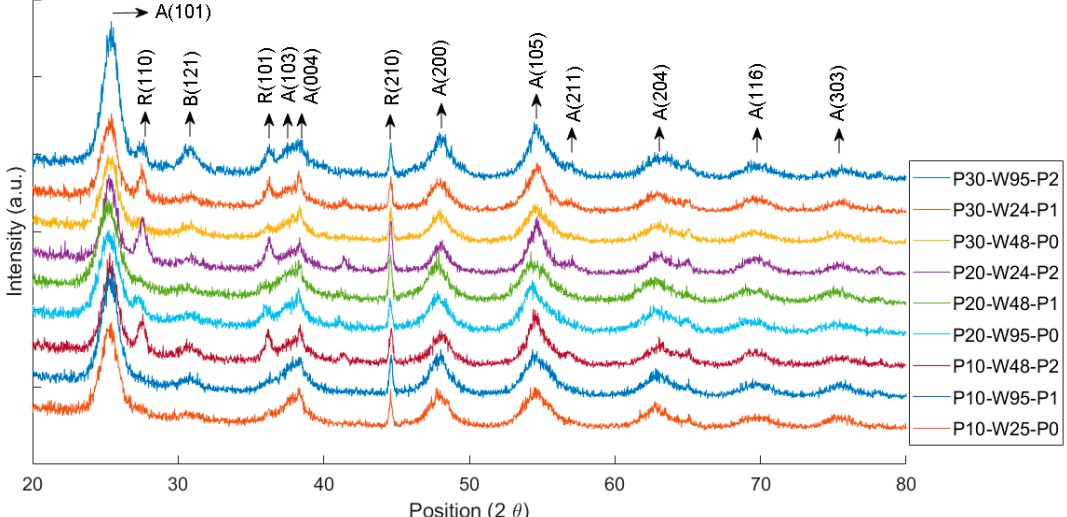

**Figure 1.** XRD pattern of samples prepared with ultrasound and calcined at 450 °C. A = anatase; R = rutile; B = brookite.

**Table 1.** Average pore size, pore volume, surface area (error within ±1%), particle size (error within ±1%), and phase fraction (error within ±6%) of the samples prepared with ultrasound and calcined at 450 °C.

| Samples | XRD | | | BET | | | PSD | |
|---|---|---|---|---|---|---|---|---|
| | Anatase % | Rutile % | Brookite % | Average Pore Size (Å) | Pore Volume (cc g$^{-1}$) | Surface Area (m$^2$ g$^{-1}$) | Median Size (μm) | Mean Size (μm) |
| TM10-49-0 | 56 | 14 | 28 | 19 | 0.13 | 101 | 9 | 10 |
| TM10-195-1 | 52 | 14 | 34 | 28 | 0.24 | 154 | 9 | 10 |
| TM10-98-2 | 45 | 20 | 35 | 24 | 0.13 | 102 | 9 | 10 |
| TM20-195-0 | 48 | 21 | 31 | 24 | 0.29 | 158 | 11 | 13 |
| TM20-98-1 | 51 | 18 | 31 | 25 | 0.20 | 132 | 7 | 8 |
| TM20-49-2 | 44 | 22 | 30 | 22 | 0.14 | 102 | 4 | 4 |
| TM30-98-0 | 47 | 18 | 30 | 28 | 0.20 | 121 | 8 | 9 |
| TM30-49-1 | 45 | 26 | 28 | 19 | 0.14 | 138 | 9 | 10 |
| TM30-195-2 | 50 | 16 | 35 | 22 | 0.25 | 128 | 9 | 10 |
| Sol-gel-450 | 86 | 7 | 7 | 19 | 0.21 | 129 | 155 | 181 |
| Sol-gel-550 | 73 | 20 | 7 | 33 | 0.14 | 48 | 160 | 183 |

Surprisingly, all of the samples synthesized with ultrasound also contain brookite in percentages around 30%. Brookite is the least-studied $TiO_2$ phase because it is very difficult to obtain it as a pure phase or in high percentages and analyze it [50]. However, DFT analysis calculated higher reactivity for the exposed brookite (210) surface than for the ubiquitous anatase (101): Li et al. reported that $H_2O$ is adsorbed on brookite (201) 30% more strongly than anatase (101) [51]. The combination of anatase and rutile has synergistic effects, compared to the pure phases. The combination of anatase and rutile in the lattice inhibits the electron-hole recombination by trapping photo-excited electrons and holes in the anatase [52]. In fact, anatase has a larger band-gap than rutile. However, the indirect band-gap of anatase is smaller than its direct one, while in the case of rutile, both are similar. For brookite, the theoretical band-gap is an intermediate between those of anatase and rutile (3.14 eV) [50]; nevertheless, its value depends on whether it is measured as a direct or an indirect band-gap, and on from the thickness of sample layers for pure crystals, reaching 3.56 eV for a direct band-gap [53]. However, semiconductors with an indirect bandgap have longer charge-carrier lifetimes than to materials with a direct bandgap. [52]. Therefore, anatase and brookite have longer electron-hole pair lives than rutile, which makes them more suitable to carry charges for longer times. Longer electron-hole pair lifetimes in anatase compared to rutile are preferable for charge carriers to participate in surface reactions [54]. On the other hand, there are many other surface properties that affect molecular adsorption and the photocatalytic activity. These include the surface morphology, the affinity of the molecules for the surface, the interaction of the molecules with the surface defects, and the surface potential, which influences the charge transfer from the photocatalyst to the molecules adsorbed [50,53,55].

## 2.2. Specific Surface Area (BET) and Pore Volume

The $N_2$ adsorption-desorption patterns of the Mn-doped $TiO_2$ powders are type IV isotherms (Figure 2) with a type H3 hysteresis loop (which indicates that powders contain mesopores from 2 nm to 50 nm). For all the catalysts, the pore size is 2–3 nm (Table 2). The isotherm type IV of the catalyst has a type E pore shape with a thinly-necked-bottle shape. The pore size decreases because (1) small crystallites aggregate and (2) Mn ions migrate into the pores of $TiO_2$ [27,50].

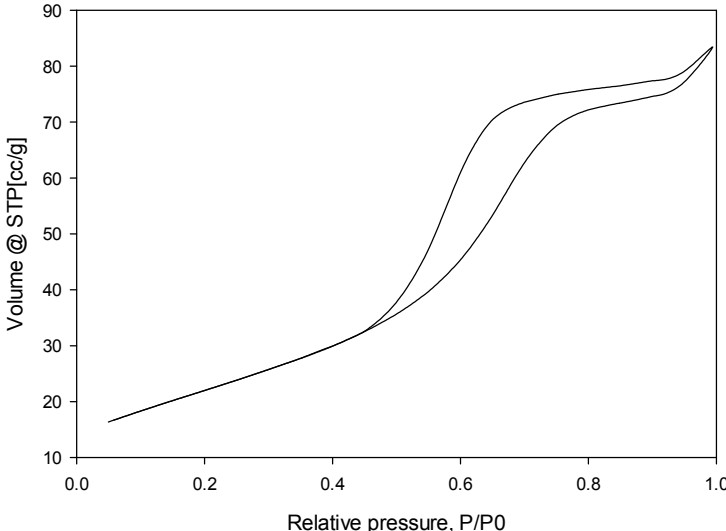

**Figure 2.** Example of N$_2$ adsorption (full points)–desorption (empty points) isotherm of a Mn-doped TiO$_2$ powder (TM-49-0).

**Table 2.** Band-gap of the samples.

| Samples | UV–Vis |
| :---: | :---: |
| | Band-Gap (eV) |
| TM10-49-0 | 1.91 |
| TM10-195-1 | 1.83 |
| TM10-98-2 | 1.91 |
| TM20-195-0 | 1.6 |
| TM20-98-1 | 1.71 |
| TM20-49-2 | 1.77 |
| TM30-98-0 | 1.65 |
| TM30-49-1 | 1.6 |
| TM30-195-2 | 1.77 |
| TM20-195-0-550 | 1.63 |
| TM30-195-2-550 | 1.65 |
| Sol-gel-450 | 1.8 |
| Sol-gel-550 | 1.72 |

*2.3. PSD*

The Mie theory applies to Mn-doped TiO$_2$ particles, as all the mean particle sizes are between 4 and 13 µm and the particle's median sizes are between 4 and 11 µm for the samples synthesized with ultrasound (Table 1). It is well known that ultrasound cavitation in comparison to the conventional sol-gel process reduces the particle size and the size of agglomerates on the support [32,33]. However, the smaller the particle, the higher the risk of sintering during calcination [31,56].

*2.4. UV–Vis Absorbance and Band-Gap*

All samples turned from white to grey during calcination. The color of the sample depends on the Mn concentration and it darkens as its concentration increases [38]. UV–Vis measured the absorbance wavelength of the samples of Mn-doped TiO$_2$ (Table 2). The absorbance wavelength of bare TiO$_2$ has a sharp edge at 400 nm, which is related to the excitation of the electrons from the valence band to the conduction band of the semiconductor (Figure 3) [29].

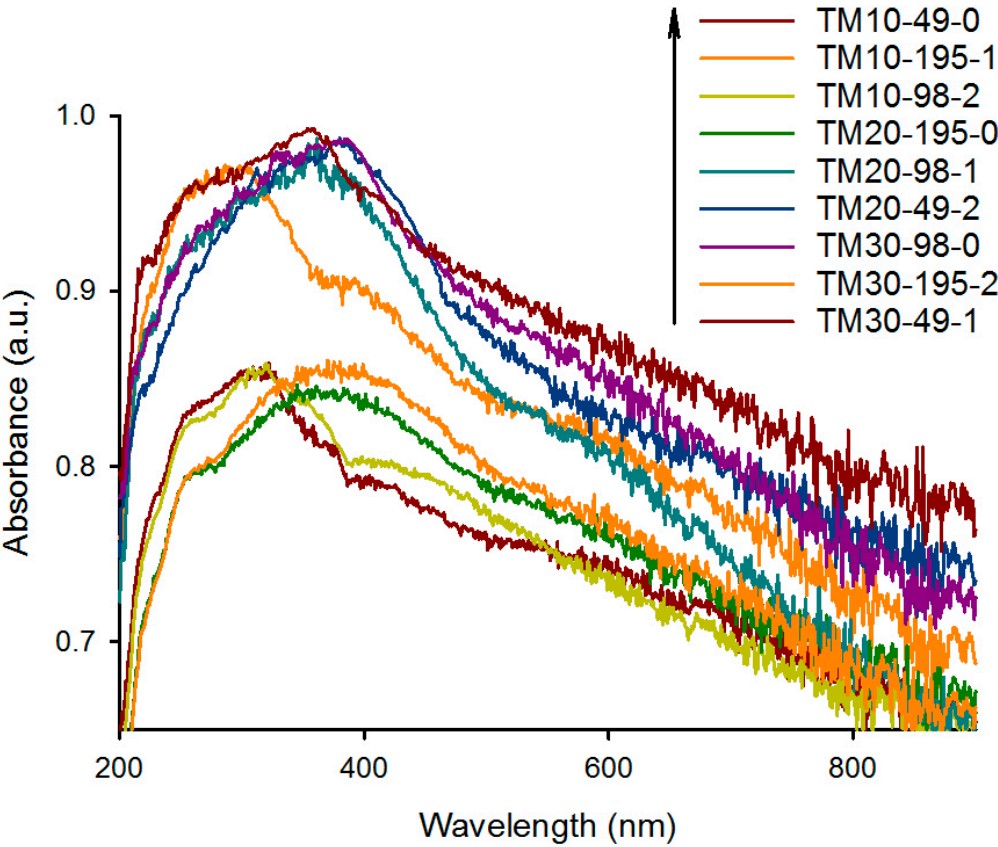

**Figure 3.** Absorbance spectra of $MnO_x$-$TiO_2$ samples.

Bare $TiO_2$ absorbs in the UV region of the spectrum as it has a band-gap energy of 3.2 eV. Metal doping red shifts the absorption of $TiO_2$ depending on the degree of doping [35,57,58]. Moreover, Aziz et al. reported that $TiO_2$ nanoparticles prepared sonochemically had a band-gap, decreasing from 3.06 eV to 2.61 eV as the ultrasound power applied decreased from 70% to 30% (maximum nominal power of 750 W). This makes the absorbance wavelength shift towards the visible region compared to the conventional sol-gel process [36,59].

The band-gap of all Mn-doped samples ranged from 1.6 eV to 1.91 eV. The shift of the absorption in the visible region is attributable to the broad absorption of some of transition metals and the effect of doping with pure $TiO_2$. Mn decreased the band-gap of the catalysts to less than 2 eV. Samples prepared at 200 W L$^{-1}$ (TM10-195-1, TM20-98-1, TM30-98-0, TM30-58-1, and TM20-58-2) have an absorbance up to 0.95 a.u. compared to the samples prepared at 150 W L$^{-1}$, whose absorbance reached a maximum of 0.85 a.u. The higher the ultrasonic power density, the more intimate the mixing among the catalyst components, leading to lower band-gap and higher absorbance.

Oxygen occupies the 2p and 4d orbitals of the valence band (VB) of $TiO_2$. DFT calculations reported the existence of oxygen vacancies in $TiO_2$ without affecting the overall band-gap that induces a donor level next to the mid-gap (deep level) defect states. The substitution of Mn ions with the lower valence and higher ionic radius $Ti^{4+}$ induces oxygen vacancies at the surface of anatase, which serves as an excellent site for $O_2$ adsorption and activation to form superoxide anion radicals ($^{\bullet}O_2{}^{-}$). By shifting the valence band to lower binding energies, the absorption peak edges shift to the red region. A high concentration of Mn provokes the recombination of electron-hole pairs, which is due to the induced lattice defects [40,56]. In a photocatalyst, when $Mn^{2+}$ traps electrons, its electronic configuration changes from $3d^5$ to $3d^6$, (Equation (1)) and when it traps holes its electronic configuration changes to $3d^4$ (Equation (3)). Both states are unstable, and to restore its stable configuration, Mn donates the

trapped electron to an oxygen molecule (Equation (2)) and the trapped hole to the water adsorbed onto the surface (Equation (4)) to generate superoxide ($O_2{}^-$) and hydroxyl ($OH^\bullet$) radicals, respectively:

$$Mn^{2+} + e^- \rightarrow Mn^+ \tag{1}$$

$$Mn^+ + O_2ads \rightarrow Mn^{2+} + O_2{}^{\bullet-} \tag{2}$$

$$Mn^{2+} + h^+ \rightarrow Mn^{3+} \tag{3}$$

$$Mn^{3+} + OH^- \rightarrow Mn^{2+} + OH^\bullet. \tag{4}$$

The half-electronic structure of $Mn^{2+}$ accelerates the charge transfer process and acts as a shallow trap for the charge carriers. Therefore, the generation of highly active oxidative species increases [47].

### 2.5. The Effect of Ultrasound Power

At low power (0–15 W), the release of energy inside the solution is insufficient to convert anatase to rutile (Figure 4, Figure 5, Figure 6, Figure 7 and Figure S1). The higher the power, the higher the amount of anatase that will turn into rutile. By increasing the amplitude, the energy within the hot-spots increases, which favors crystal growth [58]. Despite the high temperature within the hot-spots, the cooling rates are in the order of milliseconds [59]. At high amplitude, ethanol partially vaporizes, making the catalyst precursors more concentrated; thus, shifting the reaction equilibrium and increasing the phase transformation from anatase to rutile. Moreover, at power in the range of 15–20 W, the higher bulk temperature prevents gas bubbles and the energy of collapse is consequently lower. This, despite the higher bulk temperature, limits the collapse temperature and minimizes the local volatilization. At the same time, dissipated mechanical heat energy accelerates the phase transformation.

During sonication, small bubbles close to the horn coalesce and form larger bubbles. Acoustic shielding is the reflection of acoustic energy from large bubbles towards the horn, since large bubbles are incapable of absorbing acoustic energy. Because of this, the distribution of energy inside the liquid is heterogeneous. In general, at higher amplitude, acoustic shielding increases and leads to a reduction in the percentage of rutile (Table 1) [60]. At power lower than 10 W, stable cavitation occurs, whereas above 10 W, the cavitation is transient [61]. The higher the power, the higher the number of cavitation bubbles. However, there is not enough time for ethanol to accumulate at the surface of the bubble/solution and for water and ethanol to evaporate [61].

Figure 5 and Figure S2 show the effect of the power on the surface area, pore volume, and average pore size. At 20 W, which is an intermediate power, the specific surface area is the highest ($158\ m^2\ g^{-1}$), as is pore volume. At both lower and higher power than the optimum value of power, surface area and pore volume decrease, while pore size increases.

Power density, frequency, and wattage of the system define the characteristics of ultrasound [32]. A minimum intensity or power is required to achieve cavitation. Under certain conditions, particles continuously form until an optimum value of power/amplitude. High power or amplitude is necessary to achieve a sufficient mechanical vibration to promote cavitation in the sample, but high power and amplitude also deteriorate the transducer, which leads to the agitation of the solution instead of cavitation, causing poor transmission of ultrasound through the liquid medium. In addition, high power/amplitude causes undesired effects, such as the degradation of molecular structures. High power disrupts the bubble dynamic as it grows, and this leads to poor cavitation and growth of the material [62]. Additionally, at high power/amplitude, the removal efficiency of the deposit increases and there is less chance of crystallization [63]. At higher power/amplitude the cushioning effects also decrease, transferring energy out of the system, leading to lower cavitation activity [64]. Yu et al. prepared $TiO_2$ particles with ultrasound, achieving a surface area of $112\ m^2\ g^{-1}$ and a pore

size of 6.7 nm. They proved that a high amplitude creates particles with higher surface area and pore diameter (around 7 nm) [65].

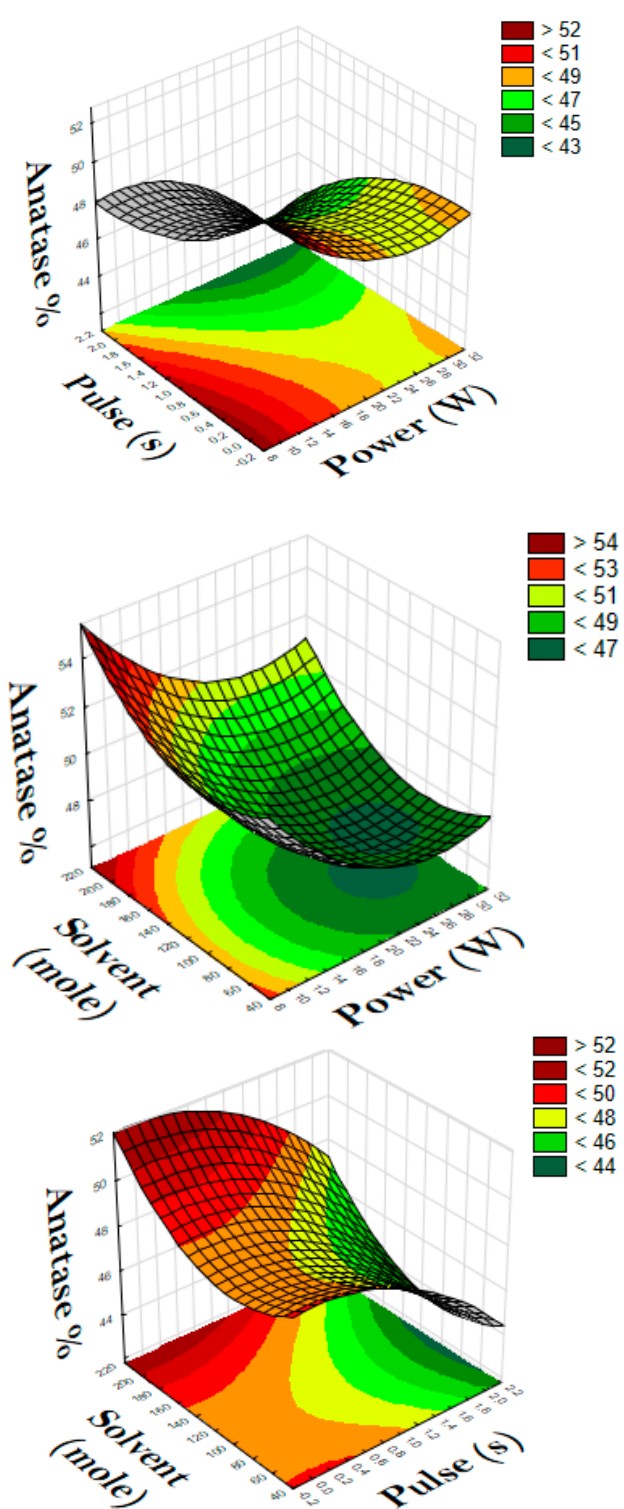

**Figure 4.** The effects of ultrasound power and pulses and the amount of solvent on anatase formation.

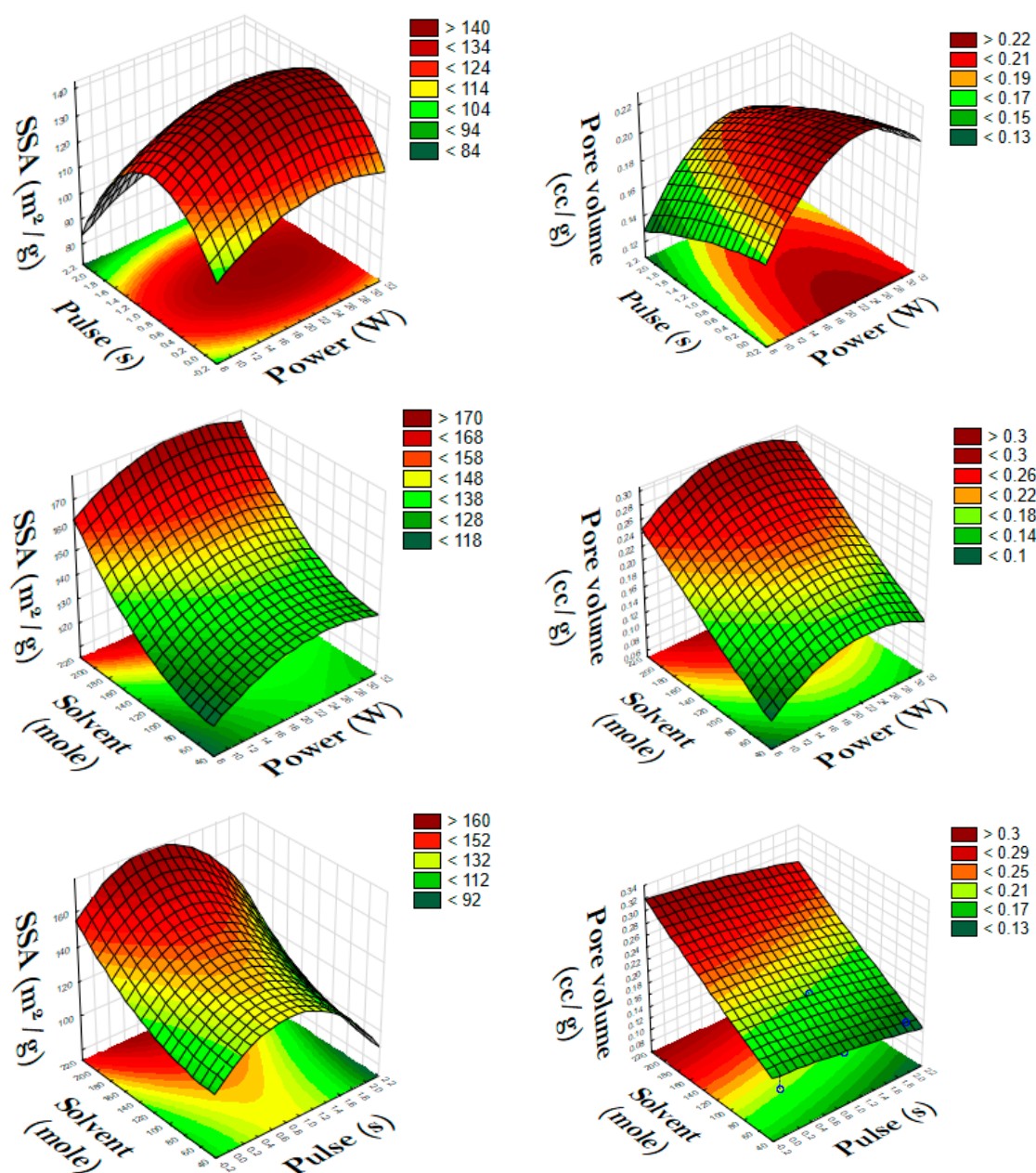

**Figure 5.** The effects of ultrasound power and pulses and solvent amount on the textural properties.

Ultrasonic power considerably affects the size of particles (Figure 6). A power of 20 W yielded the smallest particle size (4 μm), whereas with powers higher or lower than 20 W, larger particles formed (10 μm). In fact, at high power, shock waves cause particles to collide with high energy, provoking localized melting at the collision sites and initiating the agglomeration of particles. At lower power, the collision energy is not high enough to start nucleation and particles agglomerate into big clusters [63].

Figure 7 shows the effect of operating parameters on the band-gap. In general, increasing the ultrasound power results in a smaller band-gap. However, above 20 W, there is no significant effect on band-gap narrowing. Cavitation and collapsing of the bubbles induce turbulence inside the liquid and increase the diffusion of materials. Therefore, Mn diffuses more inside the $TiO_2$ lattice and replaces Ti ions, decreasing the band-gap up to a certain power, at which point the turbulence has no more effect.

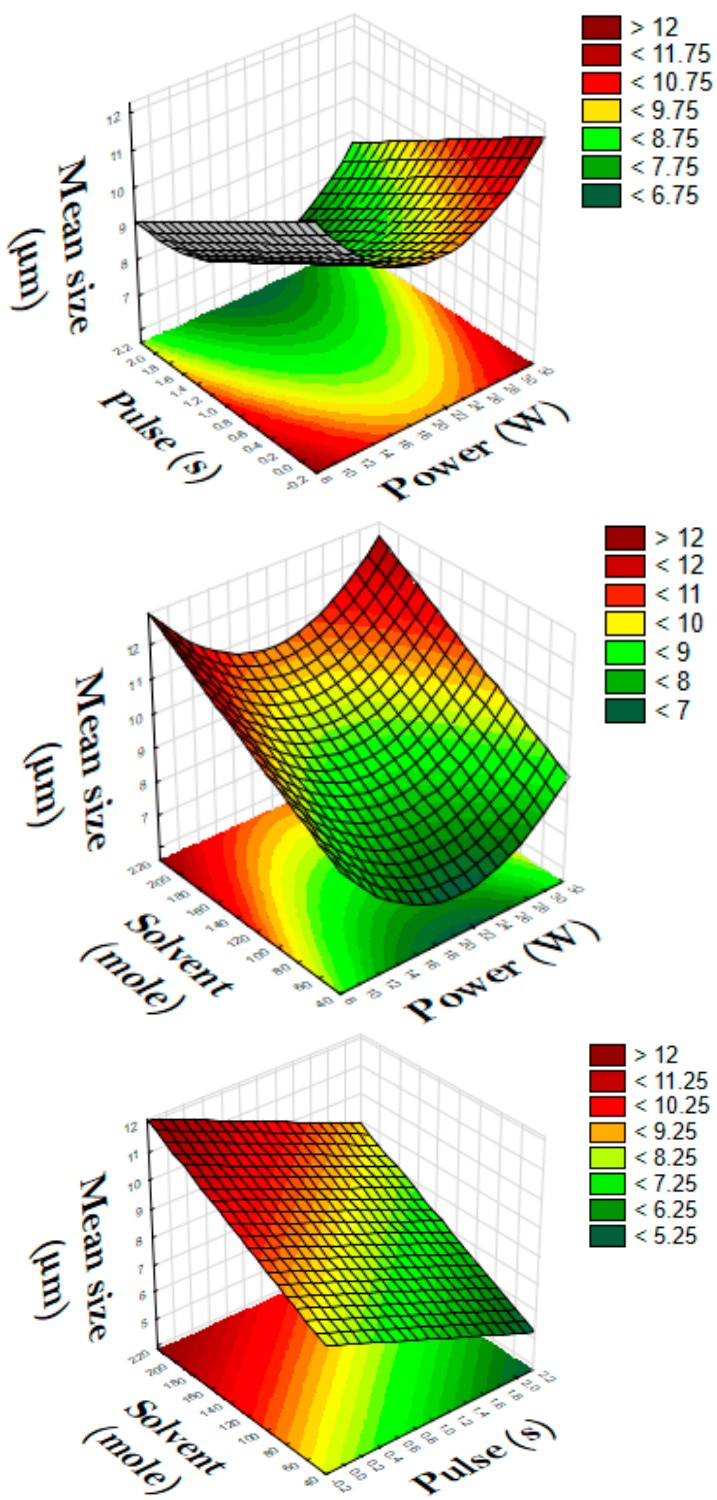

**Figure 6.** The effects of ultrasound power and pulses on particle size.

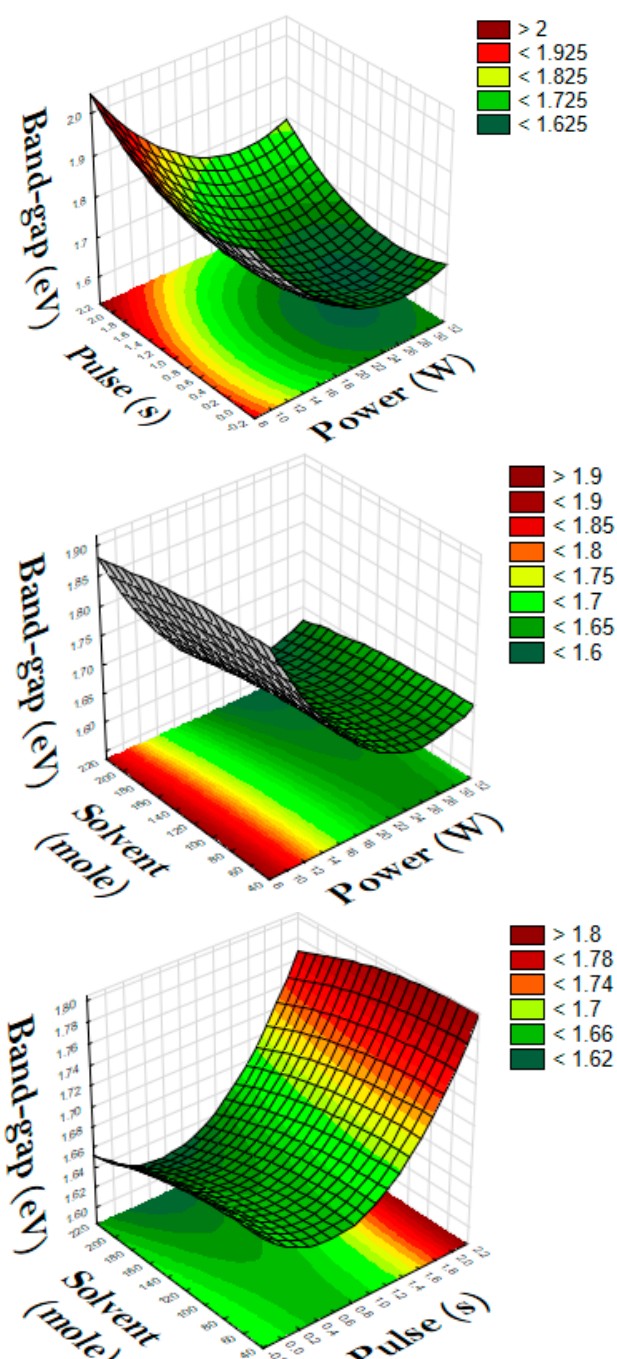

**Figure 7.** The effects of ultrasound power and pulses and the amount of solvent on band-gap.

## 2.6. The Effect of Ultrasound Pulses

To reduce the net power consumption in the system, and to cool down the transducer and investigate the effect of ultrasound pulses on the catalyst textural and optical properties, we varied the on/off ultrasound pulses ratio (Table 3). Pulse has an optimum value when it comes to the ultrasonic preparation of nanoparticles (Figure 4, Figure 5, Figure 6, Figure 7, Figure S3, and Figure S4). Pulse sonication produces smaller crystal particles. Especially for high current densities, nucleation occurs at a high rate during each pulse. In shorter pulse sonication, the growth of crystals by the creation of new nuclei, or deposition on the already existing nuclei, is slow. Crystal size may or may not increase

with the number of pulses. It depends on whether each new pulse forms a new nucleus or adds to pre-existing ones. If the new nuclei form during each pulse, the crystal size will be smaller [63].

**Table 3.** Experimental design: factorial design. US = ultrasound.

| Sample Name | Ultrasound Power (W) | Solvent (mole) | Ultrasound Pulses—On-Off (s) | Calcination Temperature (°C) |
|---|---|---|---|---|
| TM10-49-0 | 10 | 49 | Continuous US | 450 |
| TM10-195-1 | 10 | 195 | 1-1 | 450 |
| TM10-98-2 | 10 | 98 | 2-2 | 450 |
| TM20-195-0 | 20 | 195 | Continuous US | 450 |
| TM20-98-1 | 20 | 98 | 1-1 | 450 |
| TM20-49-2 | 20 | 49 | 2-2 | 450 |
| TM30-98-0 | 30 | 98 | Continuous US | 450 |
| TM30-49-1 | 30 | 49 | 1-1 | 450 |
| TM30-195-2 | 30 | 195 | 2-2 | 450 |
| Sol-gel-450 | - | 195 | No US | 450 |
| Sol-gel-550 | - | 195 | No US | 550 |

The effect of pulses can be explained in terms of times $\tau_1$ and $\tau_2$, which are characteristic of each sonochemical system pulsing. The former is the time within which ultrasound produces and then grows active gas bubbles; the latter is the time taken by all the gas bubbles present in the system to dissolve in the liquid or evaporate. For the system to activate, $\tau_1$ must be shorter than the ultrasound pulse length. When $\tau_2$ is longer than the interval time from one ultrasound pulse to the next, gas nuclei are still present in the system and the pulse that just started overlaps the previous one that is deactivating. Therefore, the active bubble population is higher with combinations of short $\tau_1$ and long $\tau_2$. Consequently, when the ultrasound-off pulse times are shorter, there is a higher probability that active bubble nuclei are still dissolved in the system [66]. This is why the SSA of the samples exhibit a clear maximum at ultrasound pulses with a 1s/1s on/off time. Moreover, at short ultrasound pulses, the ultrasound smoothly stirs the solution and crystallization has less time to occur [65,66]. On the other hand, at longer pulse sonication times, the number of active bubbles decreases due to the degassing [30]. Therefore, in order to maximize the cavitation efficiency, the optimum pulse sonication must be identified [62]. In our case, the minimum band-gap energy was achieved with an on-off pulse of 1-1 s.

The band-gap was narrower in the samples obtained with continuous ultrasound (Figure 8) and wider for longer pulses (2s/2s on/off). We explain this considering that continuous ultrasound promoted a more intimated mixing between Mn and $TiO_2$.

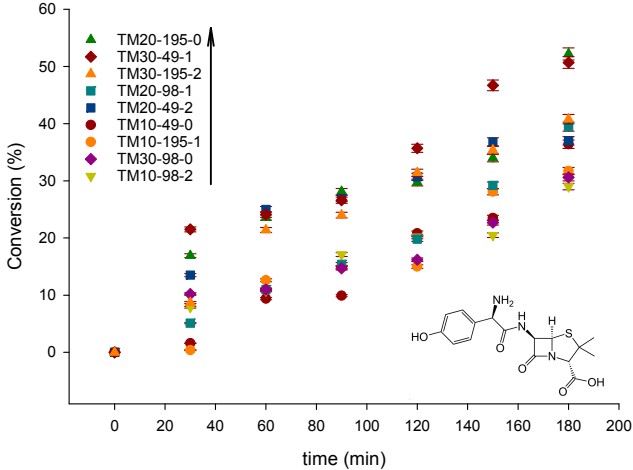

**Figure 8.** Amoxicillin degradation with prepared catalysts; error within ±2%.

## 2.7. The Effect of Solvent Volume

By decreasing the solvent volume at equal conditions of ultrasound power, the ultrasound density increases [30]. However, by decreasing the solvent volume regardless of the ultrasound power applied, the percentage of the anatase phase decreases (Figure 4). This phenomenon is attributable to the fact that ethanol suppresses the hydrolysis of titanium alkoxide and the rapid crystallization of $TiO_2$ particles, as already mentioned in the discussion about the X-ray diffractograms. Moreover, ethanol is volatile and diffuses into the cavitation bubbles. The ethanol vapor inside the bubble reduces the final temperature in the adiabatic compression. The amount of water during the sol-gel process determines the reaction mechanism and the number of active sites generated [35].

Increasing the amount of solvent results in a homogenous reaction mixture, and therefore produces particles with higher surface areas and pore volumes [33]. However, in the absence of alcohol, particles have an irregular shape. The growth of crystals occurs by increasing the amount of solvent, which, therefore, leads to particles with higher surface areas and pore volumes (Figures 4–7).

It is well known that the nucleation and the growth of crystals requires water. The higher the amount of water, the more homogeneous the nuclei and the smaller the size of particles. On the other hand, the amount of solvent changes the power density of ultrasound. Decreasing the solvent's volume increases the power density inside the liquid and prevents the agglomeration of the particles, resulting in particles of a smaller size. On the other hand, by increasing the solvent's volume, the power density inside the solution decreases and agglomeration of particles results in larger particles (Figure 6). For our samples, the amount of solvent had no significant effect on the band-gap energy (Figure 7).

## 2.8. Photocatalytic Degradation

We applied ultrasound and UV irradiation in the presence of Mn-doped $TiO_2$ powders to the aqueous solution of acetaminophen (APAP) or amoxicillin (AMO) for 3 h (Figures 8 and 9). In the degradation of AMO, the catalysts follow a very specific trend. The most active samples were TW20-195-0 and TW30-49-1, which had a combination of high surface area (158 $m^2$ $g^{-1}$ and 138 $m^2$ $g^{-1}$, respectively) and the narrowest band-gap. They degraded 53% and 51% of model pollutant, respectively. After these two samples, the most active were the catalysts with still moderately low band-gaps (1.71–1.77 eV; i.e., TM30-195-2, TM20-98-1, and TM20-49-2), which had SSAs in the 101–132 $m^2$ $g^{-1}$ range. After those, the samples with a higher band-gaps and a lower SSAs followed, having lower activities as well. The sample TM10-49-2, although having the highest band-gap (1.91 eV) and the lowest surface area (101 $m^2$ $g^{-1}$) among the samples, was an "outsider": this sample, that should have been the least active, was not. This catalyst was, however, the one with the highest anatase percentage, which may explain its activity.

In the degradation of acetaminophen (APAP), the catalysts behave differently, and do not follow any specific trend. They are also less active than in the degradation of AMO, converting a maximum of 26% of the model pollutant with the sample TM-20-98-1. The only recognizable trend has to do with the brookite percentage: the least active sample (TM10-49-0) was the one with the highest band-gap (1.91 eV), the lowest surface area (101 $m^2$ $g^{-1}$), and the lowest percentage of brookite (28%). The second least active sample (TM30-49-1), despite exhibiting a low band-gap (1.6 eV) and moderately high surface area (138 $m^2$ $g^{-1}$) also had the lowest percentage of brookite (28%). The two most active samples, TM-20-98-1 and TM20-195-0, were still samples that combined low band-gaps (1.6–1.71 eV) with high surface areas (>130 $m^2$ $g^{-1}$).

We explain the difference between the degradation of AMO and APAP with their different molecular properties, such as pKa and polar surface. Amoxicillin and acetaminophen have pKas of 2.8 and 9.4, respectively. AMO is more acidic than APAP, and therefore, it adsorbs better onto the OH-covered surface of $TiO_2$. Additionally, AMO has a polar surface area of 140 $Å^2$, while that of APAP is 49 $Å^2$ [29].

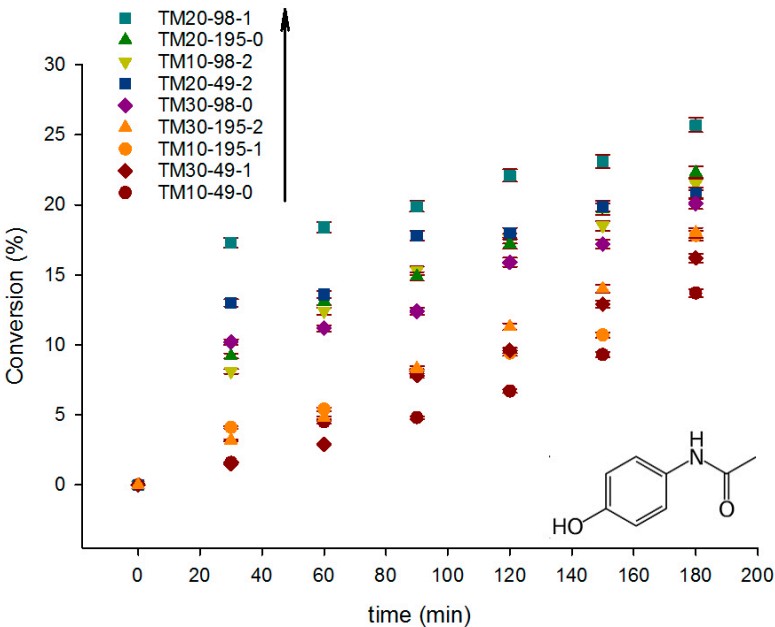

**Figure 9.** Acetaminophen degradation with prepared catalysts; error within ±2%.

Additionally, we applied ultrasound in the absence of photocatalysis for 3 h to verify the extent of degradation of ultrasound alone (Figures S5 and S6). For acetaminophen, the maximum degradation in the absence of photocatalysis and with ultrasound at 20 W reached 11.4%. Excluding the sample TM-10-49-0, which degraded 12% of the model pollutant, the degradation of acetaminophen with a combination of sonication and photocatalysis degraded significantly more acetaminophen than in the absence of a catalysts, reaching up to 26% degradation.

For amoxicillin, after 3 h, the degradation with sonication at 20 W was 29%. The three worst-performing samples did not degrade significantly more model pollutants than ultrasound alone (Figure 8). However, with sonication and photocatalysis the amoxicillin degradation reached 53%. The combination of sonication and photocatalysis processes features higher efficiency to treat wastewater when combined, rather than as individual processes.

*2.9. Comparison of the Ultrasound-Assisted Method with the Traditional Sol-Gel Method*

We prepared two samples with the conventional sol-gel process. Figure 10 shows the XRD patterns of these two samples. The fractional composition of anatase at both calcination temperatures was much higher than the sample prepared with ultrasound (Table 1). However, sol-gel samples had much less brookite (both ~7%), whereas in the samples synthesized with ultrasound the brookite content was 28% to 35%. Therefore, ultrasound affects the phase composition of the samples and this is maintained even after calcination at 450 °C. This unexpected result may open up new applications of $TiO_2$ with brookite as an active phase, which, depending on the application, is sometimes more active than anatase [50].

The particle size of the samples synthesized with traditional sol-gel is much higher than that of the samples prepared with ultrasound. The particle size of the sample obtained with the conventional sol-gel process was 181 μm, while the largest particle size of the samples obtained with ultrasound was 13 μm. Calcining the samples at 450 °C did not affect the SSA (129 $m^2$ $g^{-1}$). However, when raising the calcination temperature to 550 °C, the SSA decreased dramatically (48 $m^2$ $g^{-1}$) due to the coalescence of the pores. The pore volume also decreased along from 0.21 to 0.14 mL $g^{-1}$. The samples prepared with traditional sol-gel had a band-gap that was towards the wider end of the samples prepared with ultrasound (1.72 eV to 1.8 eV versus 1.6 eV to 1.91 eV), demonstrating that ultrasound tunes the band-gap of $MnO_x$-$TiO_2$ systems. This also opens up to exploration with ultrasound to tune

the band-gap and optical properties of other mixed oxide systems. Changing ultrasound operating parameters controls the level of mixing between the two semiconductors in the lattice and on the surface.

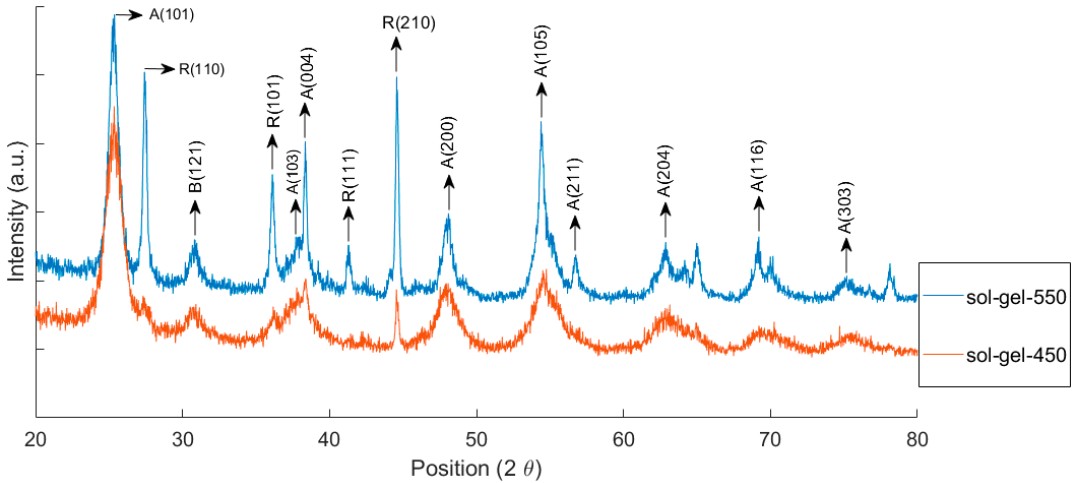

**Figure 10.** XRD pattern of samples prepared with traditional sol-gel. A = anatase; R = rutile; B = brookite.

The sample Sol-gel-450 degraded 30% of AMO in 3 h, which was comparable to the three least active catalysts synthesized with ultrasound. Indeed, this sample had a moderately high SSA (129 m$^2$ g$^{-1}$) together with a wide band-gap (1.8 eV), similarly to the least active samples.

## 3. Materials and Methods

### 3.1. Materials

Ti(IV)-butoxide (reagent grade, 97%), manganese (II) nitrate nonahydrate (reagent grade, 98%), nitric acid (ACS reagent, 70%), ethanol (HPLC grade, ≥98%), acetaminophen (APAP, analytical standard), and amoxicillin (AMO, analytical standard) were purchased from Sigma Aldrich, ON, Canada. We employed all the materials without further purification.

### 3.2. Catalysts' Syntheses

We designed our experiments based on a three-operating parameters factorial design: ultrasound power, ultrasound pulse, and solvent quantity (Table 3).

We synthesized Mn-doped TiO$_2$ catalysts by a modified sol-gel method assisted by ultrasound. A 500 W ultrasonic processor with a 20 mm tip ultrasound was powered inside the sol-gel catalyst synthesis mixture (model: VCX 500, Sonics and Materials, Inc., Newtown, CT, USA). We equipped the probe with a 20 mm diameter replaceable tip. We calibrated the ultrasonic processor periodically with the method described by Uchida and Kikuchi [67]. We added, dropwise, a solution of Mn(NO$_3$)$_2$ (1 mol) in water (24, 48, and 98 mol) to the mixture of titanium (IV) butoxide (5 mol), ethanol (25, 50, and 100 mol), and HNO$_3$ (5 mol), after starting the sonication. The reaction occurred under ultrasound irradiation at three different powers (10, 20, and 30 W) and ultrasound pulses (no pulse, 1-1 and 2-2 s on/off) for 3.5 h. The powers reported are the actual powers delivered to the system. Afterwards, the gels were aged at room temperature for 12 h and dried overnight at 100 °C in an oven under a static atmosphere. A furnace calcined the samples at 450 °C for 5 h (Table 1). We also prepared two samples by conventional sol-gel process with calcination at 450 °C and 550 °C for the sake of comparison (Table 1).

### 3.3. Catalyst Characterization

A Philips PANanalytical (USA) X'pert diffractometer measured each sample's crystallinity at ambient temperature with an angle of incidence of 0.5° and Cu K$\alpha$ (1.5406 Å) radiation at 50 kV and 40 mA. The instrument scanned the diffraction angle between 20° and 80°, with a divergence slit of 1°. We calculated the anatase, rutile, and brookite phase contents of the samples according to the Zhang and Banfield equation [68].

An AUTOSORB-1 (Quantachrome Instruments, USA) measured the specific surface area of the samples by the standard multi-point Brunauer–Emmett–Teller (BET). A furnace degassed the samples at 200 °C for 12 h in vacuum. The desorption isotherm determined the pore size distribution according to the Barret–Joyner–Halender (BJH) method with cylindrical pore size.

A laser scattering analyzer (LA-950 Horiba) determined the particle size distribution (PSD) by applying the Mie algorithm with.

A Thermoscientific UV–VIS Evolution 300 spectrophotometer equipped with a diffuse reflectance accessory (Pike technology EasiDiff) calculated the band-gap ($E_g$) of each sample with Plank's equation [29]. The absorption spectrum of each sample recorded was in the wavelength range of 200–900 nm. We calculated the bandgap with Plank's equation ($E_g = \frac{hc}{\lambda}$), where $h$ is 6.63E-34 j.c, $c$ is light speed of 3.0 E + 08 m sec$^{-1}$, and $\lambda$ is the cut-off wavelength in nm. The cut-wavelength for each absorption spectrum obtained from the intersection between the extension of the vertical section and the $h\nu$ $x$-axis.

By applying a Varian Prostar HPLC-UV instrument (model 210), equipped with a Microsorb MV 100-5 C18 column (250 mm × 4.6 mm, Variant, Agilent Technologies), we monitored the conversion of AMO and APAP. Methanol (0.5 mL min$^{-1}$) was the eluent (HPLC grade, ≥99.9%). The UV detection wavelengths were 275 nm for AMO and 210 nm for APAP.

### 3.4. Sonophotocatalytic Activity Tests

We prepared 150 mL solution of 25 ppm of APAP or 100 ppm of AMO in a jacketed glass reactor and applied continuous sonication (20 W) and UVA radiation (160 Wm$^{-2}$) for 3 h. We selected the starting concentrations of APAP and AMO based on the HPLC detection limit [18,31].

The US processor was the same as for the catalysts' syntheses. Cooling water kept the temperature of the solution at 10 °C throughout all the tests. The catalyst's concentrations were all 0.1 g L$^{-1}$.

We sampled the solution every 30 min and analyzed the products by HPLC analysis.

## 4. Conclusions

Doping TiO$_2$ with Mn increases the light absorption in the visible region due to MnO$_x$ species that absorb in the 390 nm to 730 nm range. The ultrasound-assisted synthesis of semiconductors yielded mesoporous, Mn-doped TiO$_2$ powders with a higher surface area (158 m$^2$ g$^{-1}$) and pore volume (0.29 mL g$^{-1}$), and smaller particle size (4 µm) than those obtained with the conventional sol-gel method (48–129 m$^2$ g$^{-1}$, 0.14–0.21 mL g$^{-1}$, and 181 µm, respectively). Ultrasound power and pulses, and amount of solvent (power density), elicit specific effects on the final properties of TiO$_2$ particles. In particular, moderate powers (20 W versus 10 W and 30 W), the highest solvent amount (198 mol; i.e., the lowest power density), and continuous ultrasound confer to the catalyst the desired textural (high SSA) and optical (low band-gap) properties for active photocatalysts. The catalysts degraded acetaminophen (APAP) and amoxicillin (AMO) under UV and ultrasonic irradiation. AMO decomposed more easily than APAP due to its different molecular properties (pKa and polar surface area). The maximum AMO degradation achieved was 53% with the catalyst with the smallest band-gap (1.6 eV) and the highest surface area (158 m$^2$ g$^{-1}$), whereas the maximum APAP degradation was 26% with the catalyst with the band-gap of (1.7 eV) and the surface area of (132 m$^2$ g$^{-1}$). The catalysts synthesized with ultrasound had a surprisingly high content of brookite (28% versus 7% in the traditional samples), suggesting that TiO$_2$ brookite may be at least as active as TiO$_2$ anatase in the oxidation of the model pollutants.

Ultrasound alone at 20 W degraded 11.4% and 29% of acetaminophen and amoxicillin, respectively. The worst-performing photocatalysts did not significantly degrade the model pollutants above that value. However, with the most active samples, the degradation proportions were 26% and 53% of acetaminophen and amoxicillin, respectively.

**Supplementary Materials:** The following are available online at http://www.mdpi.com/2073-4344/9/11/949/s1, Figure S1: Effect of ultrasound power on Rutile %, Figure S2: Effect of ultrasound power on average pore size, Figure S3: Effect of ultrasound pulse and solvent on Rutile %, Figure S4: Effect of ultrasound pulse and solvent on Rutile %, Figure S5: Acetaminophen degradation in the absence of photocatalysis, Figure S6: Amoxicillin degradation in the absence of photocatalysis.

**Author Contributions:** Z.K. designed and conducted the experiments, as well as analyzed the data and drafted the first version of the paper. D.S. developed the HPLC analytical techniques and helped analyzing data. D.S. also contributed substantially to the first draft of the paper. C.L.B. provided key insights on the influence of ultrasound operating parameters and the results obtained, as well improved the quality of the paper by editing it. D.C.B. conceived the study, provided insights on the characterization results of the catalysts and the photocatalytic activity results, as well as wrote and edited parts of the paper, and secured funding.

**Funding:** This research was funded by Ministère des Relations Internationales et de la Francophonie (Québec) jointly with Ministero degli Esteri (Italy), grant number QU17MO09 and by the Natural Science and Engineering Research Council (NSERC), grant number RGPIN-05628-2017.

**Acknowledgments:** The authors gratefully acknowledge the support of the Natural Sciences and Engineering Research Council of Canada (NSERC). This research was undertaken, in part, thanks to funding from the Canada Research Chairs program. The authors acknowledge Projet de coopération Québec-Italie 2017–2019 (project number: QU17MO09) for granting the mobility of researchers between Canada and Italy.

**Conflicts of Interest:** The corresponding author is co-editor of the special issue where this paper is included.

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
