# Peer review of "The Sonophotocatalytic Degradation of Pharmaceuticals in Water by MnOx-TiO2 Systems with Tuned Band-Gaps"

_catalysts, doi:10.3390/catal9110949_

Round 1

Reviewer 1 Report

Reviewer' comments

This manuscript presents a systematic study of how the textural and optical properties of Mn-doped TiO2 vary with ultrasound applied during their synthesis. Ultrasound yielded mesoporous MnOx-TiO2 powders with a higher surface area, pore volume, and smaller particle size than those obtained with the conventional sol-gel method. Authors reported that the catalysts obtained with ultrasound had a content 19 of brookite (at least 28 %), a wide distribution of the Eg. Using two pollutants(amoxicillin and acetaminophen), the higher activity was reported using Mn-TiO2 powder. This manuscript can be published with minor corrections described below;

Table 1 should be rewritten considering the proper format (all Tables).  In the line 218 of page 8, authors commented bandgap evaluation. The calculation methods have to be explained such as Kubelka-munk etc.  Fig. 3 is very strange. Did you use this absorption diagram to calculate Eg? In the Conclusion section, the last sentence can be deleted or attached to previous sentence. 

Author Response

The manuscript has been revised carefully in light of the suggestions and comments of the reviewers:

Reviewer #1:

This manuscript presents a systematic study of how the textural and optical properties of Mn-doped TiO2 vary with ultrasound applied during their synthesis. Ultrasound yielded mesoporous MnOx-TiO2 powders with a higher surface area, pore volume, and smaller particle size than those obtained with the conventional sol-gel method. Authors reported that the catalysts obtained with ultrasound had a content 19 of brookite (at least 28 %), a wide distribution of the Eg. Using two pollutants(amoxicillin and acetaminophen), the higher activity was reported using Mn-TiO2 powder. This manuscript can be published with minor corrections described below;

Table 1 should be rewritten considering the proper format (all Tables). 

Response: We thank the reviewer for this comment. We reformatted the tables in the paper (we changed the font; the size of the table and we bolded the first line in the table).

Sample name

Ultrasound power (W)

Solvent (mole)

Ultrasound pulses – on-off (s)

Calcination Temperature (°C)

TM10-49-0

10

49

Continuous US

450

TM10-195-1

10

195

1-1

450

TM10-98-2

10

98

2-2

450

TM20-195-0

20

195

Continuous US

450

TM20-98-1

20

98

1-1

450

TM20-49-2

20

49

2-2

450

TM30-98-0

30

98

Continuous US

450

TM30-49-1

30

49

1-1

450

TM30-195-2

30

195

2-2

450

Sol-gel-450

-

195

No US

450

Sol-gel-550

-

195

No US

550

Table 3. Experimental design: factorial design.

In the line 218 of page 8, authors commented bandgap evaluation. The calculation methods have to be explained such as Kubelka-munk etc.

Response: We thank the reviewer for its pertinent comment. We applied the Plank’s equation in order to measure the band gap. We added the explanation of the method that we applied.

A Thermoscientific UV–VIS Evolution 300 spectrophotometer equipped with a diffuse reflectance accessory (Pike technology EasiDiff) calculated the band-gap (Eg) of each sample with the Plank’s equation [13]. The absorption spectrum of each sample recorded in the wavelength range of 200-900 nm. We calculated the bandgap with Plank’s equation (, where h is 6.63E-34 j.c, c is light speed of 3.0 E+08 m sec-1 and λ is the cut-off wavelength in nm. The cut-wavelength for each absorption spectrum obtained with the intersection between the extension of the vertical section and the hʋ x-axis.

[1] Stucchi, M.; Elfiad, A.; Rigamonti, M.; Khan, H.; Boffito, D. C. Water Treatment: Mn-TiO 2 Synthesized by Ultrasound with Increased Aromatics Adsorption. Ultrason. Sonochem. 2018, 44, 272–279. https://doi.org/10.1016/j.ultsonch.2018.01.023

Fig. 3 is very strange. Did you use this absorption diagram to calculate Eg?

Response: We used the UV-vis absorbance to measure the absorbance vs. wavelength. We used this diagram to draw the tangent line on the straight part to the intersect of the x-axis (hυ) to have the λ cut-off wavelength in nm [1].

[1] Stucchi, M.; Elfiad, A.; Rigamonti, M.; Khan, H.; Boffito, D. C. Water Treatment: Mn-TiO 2 Synthesized by Ultrasound with Increased Aromatics Adsorption. Ultrason. Sonochem. 2018, 44, 272–279. https://doi.org/10.1016/j.ultsonch.2018.01.023

In the Conclusion section, the last sentence can be deleted or attached to previous sentence. 

Response: We thank the reviewer for its comment, we deleted the last sentence.

Reviewer 2 Report

This paper concerns the sonochemical deposition of Mn-doped TiO2, and the subsequent characterisation of outputs. The paper will likely be of interest to those in the photocatalytic degradation field and with various corrections, refinements and/or additions (including numerous formatting errors), may be considered for publication.

I don’t think the clarification between the major different phases of titania (i.e.; anatase, rutile, and brookite) is either clearly or fully treated. E.g.; full and clear outlines of indirect bandgaps; effective mass of charge carriers, variation in recombination times, etc. More broadly, I think there are far too many overly general, and insufficiently referenced, claims made throughout the manuscript. A more specific approach needs to be taken; claims need to be precise and well-referenced. A strange and confusing labelling system is used for the samples (e.g.; TM30-195-2-550). Why not clarify and clearly label all samples in graphs with a more intelligible descriptors? Section 3.1: What is the degree of accuracy/error in the brookite XRD peak fit and so, the values used in the brookite percentage calculation (i.e.; what is the degree of error)? There seems to be a huge degree of error involved from the poorly resolved peaks. Is it possible to do Raman spectroscopy on the samples? I think basing the fits and phase information of XRD data alone, especially when it is so poorly resolved, is a little shaky. Raman would help give clearer transitions and differences between the samples, and so help meaningfully reinforce the XRD data. Section 3.4: Metal doping does not always red-shift the bandgap. For example, Nb-doping (amongst others) has been shown to blue-shift. Section 3.4: The treatment/explanation of the defects states in Mn is insufficient. Defect formation energies are not constant. What is the relevant relationship for Mn in TiO2 systems? What does the data show? There is not enough of an explanation given. Section 3.8: Where is the degradation data of the two model degradants, absent the use of any catalyst? I think a reference value for comparison is required here because I would be very surprised if 3hrs of sonochemical treatment alone would not have some form of degradative effect in and of itself. The paper can be further improved by adding one or more of the following references: DOI: 10.1021/acs.jpclett.6b01501 & 10.1021/acs.jpca.5b11567 [information on charge carrier recombination rates] DOI: 10.1021/acs.chemmater.7b04944 [information on pure-phase brookite for comparison and for understanding on effect of catalytic performance] DOI: 10.1177/0040517510383618 [sol gel formed catalytic TiO2 data comparison] DOI: 10.1016/j.ultsonch.2017.06.002 [cavitation of catalytic TiO2] DOI: 10.1021/acs.jpcc.7b00290 & 10.1021/acs.jpcc.7b06715 [catalytic reduction using Mn-TiO2]

Author Response

The manuscript has been revised carefully in light of the suggestions and comments of the reviewers:

Reviewer #2:

This paper concerns the sonochemical deposition of Mn-doped TiO2, and the subsequent characterisation of outputs. The paper will likely be of interest to those in the photocatalytic degradation field and with various corrections, refinements and/or additions (including numerous formatting errors), may be considered for publication.

I don’t think the clarification between the major different phases of titania (i.e.; anatase, rutile, and brookite) is either clearly or fully treated. E.g.; full and clear outlines of indirect bandgaps; effective mass of charge carriers, variation in recombination times, etc. More broadly, I think there are far too many overly general, and insufficiently referenced, claims made throughout the manuscript. A more specific approach needs to be taken; claims need to be precise and well-referenced.

Response: We thank the reviewer for these very informative comments. We modified our paragraph to address the reviewer’s concerns more detail .  

Surprisingly, all of the samples also contain brookite in percentages around 30 %. Brookite is the least studied TiO2 phase because it is very difficult to obtain it as a pure phase or in high % and analyze it [1]. However, DFT analyses calculated higher reactivity for the exposed brookite (210) surface than for the ubiquitous anatase (101): Li et al. report that H2O is adsorbed on brookite (201) 30 % more strongly than anatase (101) [2]. The combination of anatase and rutile has synergistic effects, compared to the pure phases. The combination of anatase and rutile in the lattice inhibits the electron-hole recombination by trapping photo-excited electrons and holes in the anatase [3]. In fact, anatase has a larger band gap than rutile. However, the indirect band gap of anatase is smaller than its direct one, while in the case of rutile both are similar. For brookite, the theoretical band-gap is intermediate between those of anatase and rutile (3.14 eV) [1]; nevertheless, its value depends on whether it is measured as direct or indirect band gap, as well as from the thickness of sample layers for pure crystals, reaching 3.56 eV for direct band-gap [4]. However, semiconductors with indirect bandgap have longer charge carriers lifetimes compared to materials with direct bandgap [3]. Therefore, anatase and brookite have longer electron-hole pair life compared to rutile, which makes them more suitable to carry charges for longer times. Longer electron-hole pair lifetimes in anatase compared to rutile is preferable for charge carriers to participate in surface reactions. Longer electron-hole pair lifetimes in anatase is preferable for charge carriers to participate in surface reactions than in rutile [5]. On the other hand, there are many other surface properties that affect the molecule adsorption and the photocatalytic activity. These include the surface morphology, the affinity of the molecules for the surface, the interaction of the molecules with the surface defects, the surface potential, which influences the charge transfer from the photocatalyst to the molecules adsorbed [53, 56, 61].

A strange and confusing labelling system is used for the samples (e.g.; TM30-195-2-550). Why not clarify and clearly label all samples in graphs with a more intelligible descriptors?

Response: We thank the reviewer for pointing this out. We find that labelling the samples as we did clearly allows to identify the samples with their synthesis parameters: For instance, the sample TM10-195-1 indicates a powder made of TiO2-MnO2  synthesized with ultrasound power of 10 W, 195 moles of solvent and ultrasound pulses of 1 second on and 1 second off.

Section 3.1: What is the degree of accuracy/error in the brookite XRD peak fit and so, the values used in the brookite percentage calculation (i.e.; what is the degree of error)? There seems to be a huge degree of error involved from the poorly resolved peaks. Is it possible to do Raman spectroscopy on the samples? I think basing the fits and phase information of XRD data alone, especially when it is so poorly resolved, is a little shaky. Raman would help give clearer transitions and differences between the samples, and so help meaningfully reinforce the XRD data.

Response: We thank the reviewer for his suggestion. We added the standard deviation associated to the calculation of the phase fraction. We declared in the caption of the table. “the phase fraction sample (error within ±6)” [1].

Concerning the Raman spectroscopy, unfortunately we ran out of samples. Our department just acquired a Raman equipment, and we won’t miss to perform this type of analysis.

[1] Hengnhong, Z., and Jillian F.B., Polymorphic Transformations and Particle Coarsening in Nanocrystalline Titania Ceramic Powders and Membranes, The Journal of Physical Chemistry C, 2007, 111 (18), 6621-6629, DOI: 10.1021/jp067665

Section 3.4: Metal doping does not always red-shift the bandgap. For example, Nb-doping (amongst others) has been shown to blue-shift. Section 3.4: The treatment/explanation of the defects states in Mn is insufficient. Defect formation energies are not constant. What is the relevant relationship for Mn in TiO2 systems? What does the data show? There is not enough of an explanation given.

Response: We thank the reviewer for pointing this out. We corrected it and added a better explanation.

The band-gap of all Mn-doped samples ranged from 1.6 eV to 1.91 eV. The shift of the absorption in the visible region ascribes to the broad absorption of some of transition metals and the effect of doping of pure TiO2. Mn decreased the band-gap of the catalysts to less than 2 eV. Samples prepared at 200 W L-1 (TM10-195-1, TM20-98-1, TM30-98-0, TM30-58-1, and TM20-58-2) have an absorbance up to 0.95 a.u. compared to the samples prepared at 150 W L-1, whose absorbance reach a maximum of 0.85 a.u. The higher the ultrasonic power density, the more intimate the mixing among the catalyst components, leading to lower band-gap and higher absorbance.

Oxygen occupies the 2p and 4d orbitals of the valence band (VB) of TiO2. DFT calculations reported the existence of oxygen vacancies in TiO2 without affecting the overall band-gap that induce a donor level next to the mid-gap (deep level) defect states. The substitution of Mn ions with lower valence and higher ionic radius than Ti4+ induce oxygen vacancies at the surface of anatase which serves as an excellent site for O2 adsorption and activation to form superoxide anion radicals (O2-). By shifting the valence band to the lower binding energies, the absorption peak edges shift to the red region. A high concentration of Mn provokes the recombination of electron-hole pairs, which is due to the induced lattice defects [10],[11]. In a photocatalyst, when Mn2+ traps electrons, its electronic configuration changes from 3d5 to 3d6, (1) and when it traps holes its electronic configuration changes to 3d4 (3). Both states are unstable and, to restore its stable configuration, Mn donates the trapped electron to an oxygen molecule (2) and the trapped hole to the water adsorbed onto the surface (4) to generate superoxide (O2-) and hydroxyl (OH) radicals, respectively:

Mn2+ + e- → Mn+                                                                                                                                        (1)

Mn+ + O2ads → Mn2+ + O2-                                                                                                                     (2)

Mn2+ + h+ → Mn3+                                                                                                                                                                                                                  (3)

Mn3+ + OH- → Mn2+ + OH                                                                                                                      (4)

The half-electronic structure of Mn2+ accelerates the charge transfer process and acts as a shallow trap for the charge carriers. Therefore, the generation of highly active oxidative species increases [12].

Section 3.8: Where is the degradation data of the two model degradants, absent the use of any catalyst? I think a reference value for comparison is required here because I would be very surprised if 3hrs of sonochemical treatment alone would not have some form of degradative effect in and of itself.

Response: We thank the reviewer for its suggestion. We have carried out the UV and ultrasound (in the absence of catalyst) experiments and we observed no significant degradation of the model pollutant.

We added a sentence according to this comment.

The paper can be further improved by adding one or more of the following references: DOI: 10.1021/acs.jpclett.6b01501 & 10.1021/acs.jpca.5b11567 [information on charge carrier recombination rates] DOI: 10.1021/acs.chemmater.7b04944 [information on pure-phase brookite for comparison and for understanding on effect of catalytic performance] DOI: 10.1177/0040517510383618 [sol gel formed catalytic TiO2 data comparison] DOI: 10.1016/j.ultsonch.2017.06.002 [cavitation of catalytic TiO2] DOI: 10.1021/acs.jpcc.7b00290 & 10.1021/acs.jpcc.7b06715 [catalytic reduction using Mn-TiO2]

Response: We thank the reviewer for its suggestion. Two of the references where helpful in the explanation of section 2.1 and section 2.3.

Round 2

Reviewer 2 Report

Whilst most of the changes are sufficient, I think the catalytic degradation data still needs work. There needs to be clear addition of the actual values to the broader dataset. A single sentence stating minimal change is not enough. Nothing should be expected to be taken on trust. The raw data, placed in the context of he full sequence of data, is required.  Reference 41, 51, 56, 57 contain formatting errors. Needs correction.

Author Response

1) We thank the reviewer for his comment. We are sorry as we did not completely get what he meant in the first rounds of comments. We performed the tests with only ultrasound (20 W) and in the absence of a photocatalyst. We provide the graphs in the supporting information, while commenting the results in the manuscript as follow:

Additionally, we applied ultrasound in the absence of photocatalysis for 3 h to verify the extent of degradation of ultrasound alone (Figure S-5, S-6). For acetaminophen the maximum degradation in the absence of photocatalysis and with ultrasound  at 20 W reached 11.4 %. Excluding the sample TM-10-49-0, which degraded 12 % of the model pollutant, the degradation of acetaminophen with a combination of sonication and photocatalysis degraded significantly more acetaminophen than in the absence of  a catalysts, reaching up to 25% degradation.

For amoxicillin,  after 3 h the degradation with sonication at 20 W was 29%. The three least performing samples do not degrade significantly more model pollutants than ultrasound alone (Figure 8). However, with sonication and photocatalysis the amoxicillin degradation reached 52%. The combination of sonication and photocatalysis processes features higher efficiency to treat wastewater when combined rather than as individual processes.

In the conclusions, we also addedded the following sentence:

Ultrasound alone at 20 W degraded 11.4 % and 29 % of acetaminophen and amoxicillin, respectively. The least performing photocalysts did not significantly degrade the model pollutants above this value. However, with the most active samples, the degradation was 25 % and 53 % of acetaminophen and amoxicillin, respectively.

2) We corrected the references. We thank you the reviewer for the detailed revision provided.

Round 3

Reviewer 2 Report

Changes/additions are fine. 

One final change. The newly added paragraph to the conclusions is helpful. However, it directly contradicts the preceding paragraph. In one you mention 25% and 53%, in the other 26% and 52%. Please address this inconsistency. Apart from that, all else seems fine. 

Author Response

We thank the reviewer for pointing this mistake out. We corrected it.

Regards,